# Developing an Efficient Processing System Treatment for the High Concentration of Eucalyptus Chemical Mechanical Pulp Wastewater

**DOI:** 10.3390/molecules27185774

**Published:** 2022-09-07

**Authors:** Zaiheng Huang, Xiang Qin, Tian Zhu, Xiang Yu, Mengyu Liu, Guangzai Nong, Qifeng Yang, Shuangfei Wang

**Affiliations:** 1School of Resources, Environment and Materials, Guangxi University, Nanning 530004, China; 2Guangxi Bossco Environmental Protection Technology Company Limited, Nanning 530007, China; 3School of Light Industry and Food Engineering, Guangxi University, Nanning 530004, China; 4Guangxi Key Laboratory of Clean Pulp & Papermaking and Pollution Control, Guangxi University, Nanning 530004, China

**Keywords:** chlorine dioxide, wastewater, treatment, DPAT treatment

## Abstract

The current wastewater treatment method shows low efficiency in treating wastewater with high concentrations of chemical mechanical pulp (CMP). Therefore, a chlorine dioxide Pretreatment Anaerobic Treatment (DPAT) was developed and applied to treat the CMP wastewater to obtain higher efficiency, obtaining the following results: The biodegradability of CMP wastewater improved after chlorine dioxide pretreatment. The COD of wastewater treated with chlorine dioxide was reduced from 5634 mg/L to 660 mg/L. The removal rate for chemical oxygen demand (COD) was 88.29%, 29.13% higher than the common anaerobic treatment. The reasons for the high efficiency of the DPAT treatment were that chlorine dioxide pretreatment removed the toxic substances in the original wastewater and thereby promoted the proliferation and growth of the anaerobe. The results show that pretreatment with chlorine dioxide can effectively enhance the biodegradability of high-concentration CMP wastewater. Therefore, DPAT treatment of high-concentration CMP wastewater is beneficial to environmental protection.

## 1. Introduction

The continuous development of society has led to the constant expansion of the paper industry. In the paper industry, a chemical mechanical pulp (CMP) is prepared for wrapping papers, generating a large amount of wastewater. This causes the water to have a high concentration and color, containing carbohydrates, lignin, and their degradation products [1,2]. As most of these components are difficult for microorganisms to digest, anaerobic microbial treatments have little effect [3,4,5].

Therefore, the problem of effectively treating high-concentration biomass wastewater has been a significant issue. For example, Tian et al. prepared a new ZnAl_2_O_4_/Bi_2_MoO_6_ composite material to treat an industrial eucalyptus base CMP wastewater, obtaining a COD removal rate of 50.88% [6]. Priyadarshinee et al. found that the lignin in eucalyptus-based CMP wastewater can be treated by microorganisms and, therefore, significantly reduce the COD value of the CMP wastewater [7]. D. Xing et al. treated a kind of CMP wastewater with calcification and realized its alkali-recycling applications, saving the costs needed to bay sodium hydroxide [8]. However, this research has a limited following: the COD removal rate by the ZnAl_2_O_4_/Bi_2_MoO_6_ composite material was low, at only 50.88%. It took a long time to extract lignin from CMP wastewater using microorganisms, which led to inefficient industrial production. In addition, calcium lignin waste solids were generated by treating CMP wastewater with calcification.

In addition to the above studies, Tran, Jung, and Pawar studied wastewater treatment using microbial fuel cells (MFCs), microbial sediment fuel cells (SMFCs), and microbial electrolysis cells (MECs), respectively, which are emerging bioenergy-based raw materials [9,10,11]. Wastewater treatment technology has excellent prospects, but it is not widely used. Hence, it is necessary to improve the effectiveness of CMP wastewater treatment.

Chlorine dioxide is a strong oxidizer used to treat high-concentration biomass wastewater. For example, Jin et al. studied the oxidative degradation of phenol-containing wastewater using chlorine dioxide, where the COD removal rate was over 90% [12]. Wang et al. degraded 2-sec-butyl-4,6-dinitrophenol (DNBP) production wastewater using chlorine dioxide oxidation. Decolorization and COD removal rates of 79.3% and 68.5% were obtained, respectively [13]. These pieces of research demonstrated that chlorine dioxide could be used to treat high concentrations of organic wastewater. Therefore, it should have the potential for use as a material to treat CMP wastewater.

However, the removal rate of COD when using only chlorine dioxide is still low. Therefore, combining this with an anaerobic treatment unit is better to increase the COD removal rate further. Hence, a DPAT treatment was developed to treat the high concentration of eucalyptus CMP wastewater in this paper. As a result, a good COD removal effect was obtained, and DPAT treatment was proposed to have a reaction mechanism. Therefore, this demonstrates that the developed DPAT treatment is effective for high concentrations of biomass wastewater. This research benefits the scientific theory of wastewater treatment for environmental protection.

## 2. Methods and Materials

### 2.1. Material

High-concentration eucalyptus CMP wastewater, ozone-treated CMP wastewater, and anaerobic granular sludge were all taken from Guangxi Bossco Environmental Protection Technology Co., Ltd. (Nanning, China). The COD value of the wastewater was 5634 mg/L. The initial physicochemical properties are shown in Table 1. Drugs A and B for chlorine dioxide disinfectant were bought from Guangxi Aikening Disinfection Technology Co., Ltd. (Nanning, China). A is stable chlorine dioxide, B is a supporting activator, and the main component is acid.

### 2.2. Reactor

The DPAT treatment applied in this research mainly consisted of a reactor, a flask, and a circulating pump, as shown in Figure 1. The main body of the reactor was made of hard plastics with a volume of 1.0 L, while the reactor was placed at a constant temperature of 37 °C. There was a water inlet with a diameter of 30 mm at the bottom of the reactor and a water outlet with a diameter of 30 mm above the side wall of the reactor. The circulating pump pumped out the CMP wastewater sample placed in the flask to enter the bottom of the reactor through the water inlet. The CMP wastewater was filled into the inside of the reactor, where the anaerobic granular sludge catalyzed degradation reactions. In this way, it formed a circulating water system to prevent its thermocatalytic effects.

### 2.3. Preparation of Chlorine Dioxide

The chlorine dioxide disinfectant was prepared by two used drugs, A and B, as shown in Figure 2 [14].

As shown in Figure 2, drugs A and B were fully dissolved in water. Then, the two solutions were mixed, covered, and left to stand for 30 min to obtain a mother liquor, diluted with water at a ratio of 1:10 to obtain a chlorine dioxide disinfectant.

### 2.4. Investigation of the Concentration of Chlorine Dioxide Disinfectant

Chlorine dioxide was detected by titration by sodium thiosulfate, using iodometry as an indicator. At the same time, a blank sample was set as a comparison to exclude the possible interference of chlorine in the water [15].

The concentration of chlorine dioxide was calculated by Equation (1).
(1)$$X=\frac{\left(V\;-{\;V}_{0}\right)\times \;C\;\times 0.01349}{m}\times 10^{6}$$
where V is the volume of sodium thiosulfate standard solution consumed by titration; V_0_ is the volume of the standard solution of sodium thiosulfate consumed in the blank titration experiment; C is the concentration of sodium thiosulfate standard titration solution; m is the mass of the stock solution; 0.01349 is the mass of chlorine dioxide in grams, equivalent to 1 mL of standard titration solution of sodium thiosulfate [16].

### 2.5. Processing Process

#### 2.5.1. Chlorine Dioxide Pretreatment

The 300 mL high-concentration eucalyptus CMP wastewater was placed in a flask without any pretreatment, and then 20–60 mL of chlorine dioxide disinfectant was added for pretreatment for 30 min. At the same time, a comparison experiment was carried out using an ozone oxidizer.

#### 2.5.2. Ozone Pretreatment

The 300 mL high-concentration eucalyptus CMP wastewater was placed in a vented bottle without any treatment. An oxygen bottle and an ozone generator were connected, and a certain amount of ozone gas was injected into the ventilation bottle for the reaction.

#### 2.5.3. Anaerobic Biological Treatment

A total of 2.0 g of the wet granular sludge was placed into the anaerobic reaction device. Then, 30 mL of the eucalyptus CMP wastewater was added by chlorine dioxide pretreatment. Consequently, the cycling pump started and ran for 2 to 5 days. The treated wastewater was poured out at a particular time during the treatment process to obtain the treated wastewater samples for further testing.

### 2.6. Analysis of Processing Results

The processing results were assessed based on the COD and BOD removal rates. The samples for COD values were analyzed using the potassium dichromate method, while the samples for biochemical oxygen demand (BOD) values were tested using the dilution and inoculation method. Microwave digestion instrument WXJ-III and biochemical incubator LRH-250 were used.

### 2.7. Mechanism Analysis

#### 2.7.1. GC–MS Detection

A total of 10 mol·L^−1^ sulfuric acid was added to 500 mL of CMP wastewater to acidify the solution for precipitate lignin. Then, the generated mixtures were separated by filtration to obtain a lignin-free filtrate, while the lignin precipitates were removed. After that, the obtained lignin-free filtrate was extracted three times with 50 mL of ether each time to obtain the extracted solution, which was transferred to a stoppered conical flask. The seldom water contained in the ether extracted solution was removed to obtain its dry matter using a small amount of anhydrous sodium sulfate. Finally, the dried ether extracted solution was concentrated to about 1.0 mL, which acted as the sample for the 5973 GC–MS analyzer.

The detection conditions of the instruments: chromatographic column HP5MS quartz capillary column, with a column length of 30 m, column inner diameter of 0.25 mm, and film thickness of 0.25 μm. Chromatographic separation conditions: column temperature 40 °C inlet temperature from 250 to 300 °C; vaporization temperature 280 °C; carrier gas by He gas in the flow rate of 1.0 mL·min^−1^. Split ratio 50:1, injection volume 1 μL. Mass detector: EI source, electron energy 70 eV, source temperature 230 °C.

#### 2.7.2. Microbiological Analysis

The diversity of microorganisms in the sludge in anaerobic biological treatment systems was assessed using Alpha diversity. The microorganism adaptation was evaluated by detecting the abundance of the microbial communities.

#### 2.7.3. Investigation of the Reaction Mechanisms of COD Removal by DPAT Treatment

The reaction mechanisms of COD removal using DPAT treatment were investigated based on the organic matter compositions before and after pretreatment by chlorine dioxide, the organic matter compositions before and after the following anaerobic treatment, and the abundance and diversity of the microorganism community in anaerobic treatment.

## 3. Results and Discussion

### 3.1. Result of Generation of the Chlorine Dioxide Disinfectant

The chlorine dioxide disinfectant was presented at a 465.41 mg/L concentration. The chlorine dioxide disinfection solution titrated consumed 0.7 mL of sodium thiosulfate standard solution; the blank titration sample consumed 0.01 mL of sodium thiosulfate standard solution, namely, V = 0.7 mL, V_0_ = 0.01 mL, C = 0.1 mol/L, and m = 2 g. Substituting the concentration of chlorine dioxide disinfectant into Equation (1) obtains 465.41 mg/L.

### 3.2. Results of Chlorine Dioxide Pretreatment

#### 3.2.1. The Change in COD and BOD_5_ Values with Chlorine Dioxide Treatment

The COD and BOD_5_ values of the CMP wastewater that underwent chlorine dioxide pretreatment were tested, as shown in Figure 3. The COD values of the CMP wastewater treated with chlorine dioxide were approximately 3500–4400 mg/L. With the increase in chlorine dioxide dosage, COD gradually decreased. In contrast, the BOD_5_ values of the pretreated CMP wastewater were approximately 500–910 mg/L under the condition of from 20 to 40 mL of the chlorine dioxide disinfectant applied. However, the BOD_5_ values sharply decreased to 12–16 mg/L when 50–60 mL of the chlorine dioxide disinfectant was used. Therefore, based on a COD value of 5634 mg/L for the initial CMP wastewater, the COD removal rates ranged from 21.5% to 37.3% following chlorine dioxide pretreatment.

#### 3.2.2. The Change in the Organic Contaminants by Chlorine Dioxide Treatment

A GC–MS system detected the compositions and relative levels of organic contaminants. Table 2 shows the results of the GC–MS analysis of eucalyptus CMP wastewater after chlorine dioxide treatment.

Almost all the organic compounds in the wastewater were significantly removed and converted to 3-octanone, DL-lipoamide, L-lactide, and other substances that the GC–MS analysis system could not detect. Therefore, most toxic substances such as caproamide were removed entirely, which should benefit the anaerobic treatment in the next step.

#### 3.2.3. The Ratio between the Masses of COD and Chlorine Dioxide Applied

The COD mass can be calculated by Equation (2). and the mass of the applied chlorine dioxide can be calculated by Equation (3). The ratio between the COD mass and the mass of the applied chlorine dioxide can be calculated, as shown in Table 2.
(2)$$M_{\mathrm{COD}}=C_{\mathrm{COD}}\times V_{\mathrm{WW}}$$
(3)$$M_{\mathrm{CD}}=C_{\mathrm{CD}}\times V_{\mathrm{CD}}$$
where M_COD_ is the COD mass, C_COD_ is the concentration of COD in the water sample, V_WW_ is the volume of the water sample, M_CD_ is the mass of chlorine dioxide, C_CD_ is the concentration of chlorine dioxide disinfectant, and V_CD_ is the volume of the applied chlorine dioxide disinfectant.

Table 3 shows that the ratio between M_COD_ and M_CD_ was approximately 142.46–43.19 in the chlorine dioxide pretreatment stage. This means that 1.0 g of chlorine dioxide should be used to treat 43.19–142.46 g of COD mass in the wastewater. When 40 mL of chlorine dioxide disinfectant was applied, the ratio was 60.85, implying that 1.0 g of chlorine dioxide was needed to treat 60.85 g of COD mass. This ratio resulted in the lowest COD value in the pretreatment stage.

### 3.3. The Result of Anaerobic Treatment

#### 3.3.1. The COD Removal Rate

The results of anaerobic treatment were based on the COD values and COD removal rate, calculated by Equation (4):(4)$$r=\left(1-\frac{c_{0}-c_{1}}{c_{0}}\right)\times 100\%$$
where c_0_ is the initial COD concentration, and c_1_ is the COD concentration at a specific time, mg/L.

Figure 4 shows the variation trend of COD values and their removal rates using a comparison between DPAT treatment and the common anaerobic treatment. The COD removal rate with DPAT treatment reached a maximum value of 88.29%, which was 21.27% higher than the 67.02% rate obtained by the common anaerobic treatment system [17]. Anaerobic efficiency reached 80.10% when using DPAT treatment, while the anaerobic efficiency of the common anaerobic treatment system only reached 67.02%. Therefore, treating CMP wastewater with DPAT treatment should significantly improve the efficiency of CMP wastewater treatment and should reduce the work needed in subsequent advanced treatments.

#### 3.3.2. The Change in Organic Compositions by Anaerobic Treatment

Table 4 shows the chemical species and relative content percentages of organic compounds before and after the anaerobic treatment of eucalyptus CMP wastewater using chlorine dioxide pretreatment. The relative contents of 3-octanone, DL- lipoamide, and L-lactide were significantly reduced, by 86.96%, 100.00%, and 96.23%, respectively. At the same time, 1,2-propanediamine and 2-methyl-2-butanol were present at low levels.

As many organic pollutants were removed in the pretreatment step, the water environment was more suitable for the reproduction and growth of microorganisms. Therefore, the degradation efficiency of the electromechanical slurry was higher in the anaerobic stage. Hence, it can be concluded that the eucalyptus CMP wastewater treated with DPAT treatment was more suitable for anaerobic processing and should significantly improve the anaerobic efficiency.

### 3.4. Result of Microbiological Analysis

#### 3.4.1. Microbial Abundance

To further analyze the feedback mechanism of DPAT treatment, a high-throughput 16S rRNA sequencing technology was applied to reveal the microbial abundance evaluated by the Chao1 index and Ace index, as shown in Figure 5. The Chao1 and Ace indices represent microbial abundance, the values of which are expected to be high [18]. A richer microbial abundance implies that the treated wastewater is more fit to bleed and grow and that the rich microorganisms should quickly degrade the pollutants in the wastewater.

As shown in Figure 5a,b, the Chao1 index and Aceindex in the DPAT treated wastewater were higher than those obtained using a common anaerobic treatment and an anaerobic treatment with ozone pretreatment. This demonstrated that the microbial abundance obtained by the DPAT method was higher than that of the two control methods.

#### 3.4.2. Microbial Diversities

The microbial diversities activated in sludge after a common anaerobic treatment were analyzed by a high-throughput 16S rRNA sequencing technology and evaluated by Shannon and Simpson indices, as shown in Figure 6. The Shannon value should have a higher value, as a higher Shannon number means more different microorganisms are present. However, the Simpson index values should be reduced, as a lower Simpson index implies a low possibility of the same microorganisms being present at two different testing times. Therefore, a lower Simpson index suggests a more diverse microorganism community [19,20].

As shown in Figure 6, the Shannon index was higher than those of the two control methods, while the Simpson index was significantly lower than those of the two control methods. This indicated a higher level of community diversity in the sludge activated by DPAT treatment.

#### 3.4.3. The Compositions of Microorganisms in Their Community

Figure 7 shows the changes in the phylum-level microbial diversity during the different anaerobic treatment conditions. The sludge from the CMP wastewater treated by a common anaerobic treatment showed Chloroflexi, Firmicutes, and Bacteroidetes proportions of 43.98%, 27.23%, and 10.19%, respectively. These proportions were 47.07%, 17.15%, and 15.16%, respectively, in the sludge from the CMP wastewater that underwent anaerobic treatment with ozone pretreatment. The dominant flora were Chloroflexi, Firmicutes, Bacteroidetes, and Proteobacteria, which formed 41.15%, 15.57%, 16.37%, and 10.88% of the DPAT treatment sludge, respectively. This evidenced a comparative increase in microbial diversity in the sludge activated by DPAT treatment [21]. An increase in microbial diversity is conducive to flora growth; hence, this improves the efficiency of anaerobic treatment [22].

### 3.5. The Reaction Mechanisms of a DPAT Treatment

According to the oxidation properties of chlorine dioxide and the reaction properties of the anaerobic treatment, combined with the abundance and diversity of the community of microorganisms, the reaction mechanisms for the DPAT treatment of high-concentration eucalyptus CMP wastewater included five steps, as shown in Figure 8.

In the DPAT treatment, ClO_2_ decomposes to generate peroxide and oxygen and chlorine oxidant free radicals [23].Then, the generated peroxide further decomposes to generate hydroxyl free radicals [24].The free radical electrophilically attacks the pollutants in the CMP wastewater, leading to a change in the toxic substances. This generates some new, non-toxic substances [25,26,27]. The GC–MS detection results supported this adjustment.The generation of non-toxic substances led to the reproduction and growth of the microorganisms. This adjustment is supported by the abundance and microbial diversity mentioned in the following.The growing microorganisms further degraded the non-toxic substances, resulting in a higher COD removal rate. At the same time, the reactions generated CO_2_, H_2,_ and water [28]. This adjustment was supported by the final COD removal rate, the abundance, the microbial diversity, and the improvement in the efficiency during the anaerobic reaction stage.

Chloroflexi is an anaerobic multicellular filamentous microorganism that can generate H_2_ by degrading organic pollutants in water to provide methanogens. Therefore, reducing the inhibitory effect of organic matter can ensure the stable progress of the anaerobic reaction [29]. Firmicutes have high-strength cell walls, which can stably provide proteases, cellulases, lipases, and other extracellular enzymes for the anaerobic system. They are closely related to the degradation of organic matter [30]. In an anaerobic system, various microorganisms act synergistically and promote each other to achieve the degradation effect of pollutants.

In summary, due to the oxidant reaction caused by chlorine dioxide, the original toxic pollutants in CMP wastewater are changed into some non-toxic substances. This improves the reproduction and growth of microorganisms, resulting in increased microbial abundance and community diversity. Furthermore, the high microbial abundance and diversity can significantly improve the anaerobic efficiency.

### 3.6. The Advance and Shortage of This Research

The advantages and shortcomings of the DPAT treatment can be summarized as follows: (1) DPAT treatment was developed to treat high concentrations of eucalyptus CMP wastewater. It presented a higher COD removal rate than common anaerobic treatment and anaerobic treatment with ozone pretreatment. (2) This treatment is more suitable for increasing plant processing efficiency than the existing methods. (3) The mechanisms of DPAT treatment were revealed. According to the oxidation properties of chlorine dioxide, the reaction properties of the anaerobic treatment, and the abundance and diversity of the community of microorganisms. (4) Although the COD removal rate was high, a relatively high COD value remained in the residue wastewater. Therefore, follow-up treatments are required to further remove the remaining COD values and meet the standard discharge revels [31]. (5) In addition, some toxic gas is generated in the chlorine dioxide preparation process, which is harmful to health [32].

## 4. Conclusions

The following conclusions were drawn through experiments and analysis: A DPAT treatment was successfully developed to treat high concentrations of eucalyptus CMP wastewater, obtaining the maximum COD removal rate of 88.29%. COD removal rates were 29.13% higher than those obtained with common anaerobic treatment. As the original toxic pollutants in the CMP wastewater charged to some non-toxic substance, the microbial abundance and diversity increased in the anaerobic treatment, significantly improving the anaerobic efficiency.

## Figures and Tables

**Figure 1 molecules-27-05774-f001:**
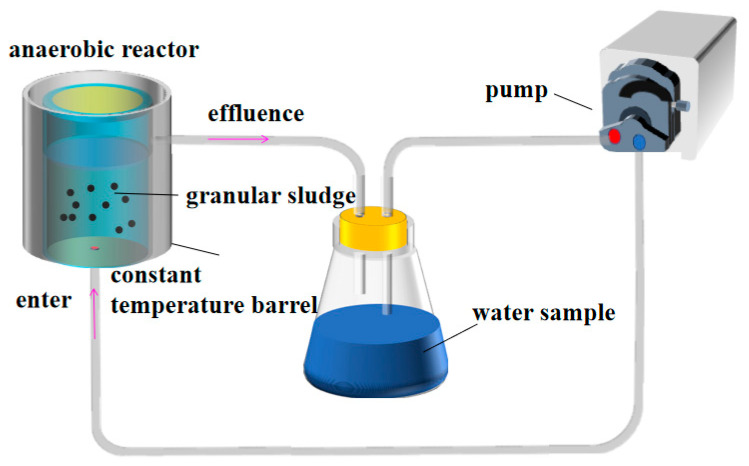
Anaerobic reaction device.

**Figure 2 molecules-27-05774-f002:**
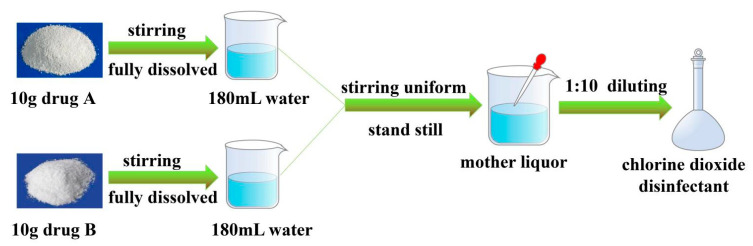
The preparation process of chlorine dioxide disinfectant.

**Figure 3 molecules-27-05774-f003:**
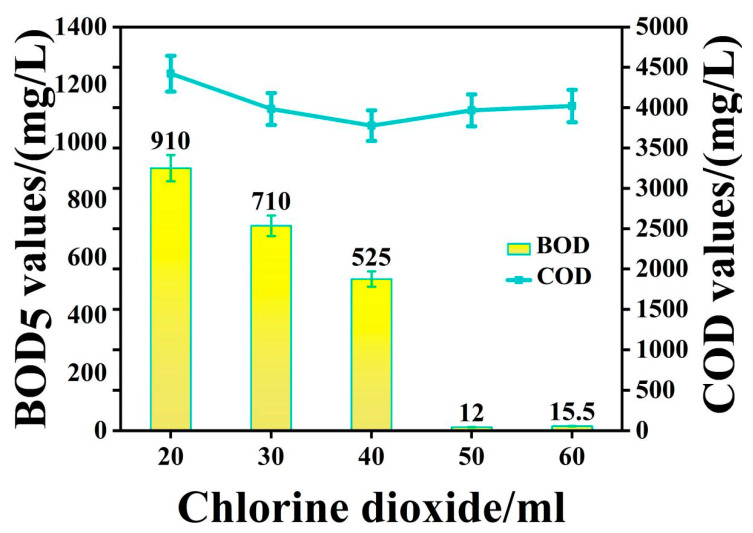
The BOD_5_/COD values after chlorine dioxide pretreatment.

**Figure 4 molecules-27-05774-f004:**
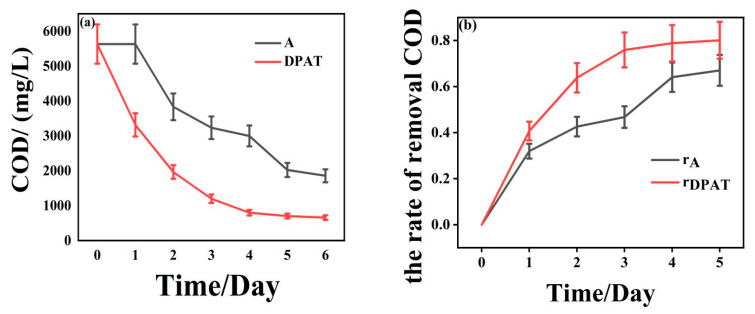
The change in COD values (**a**) and their removal rates (**b**). A and DPAT represent the variation trend of COD under different systems. r_A_ and r_DPAT_ represent the changing trend of COD removal rate under different systems.

**Figure 5 molecules-27-05774-f005:**
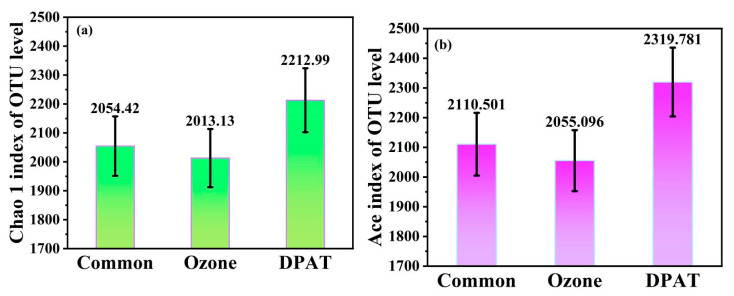
The microbial abundance in the activated sludge samples evaluated by the chao1 index (**a**) and the ace index (**b**).

**Figure 6 molecules-27-05774-f006:**
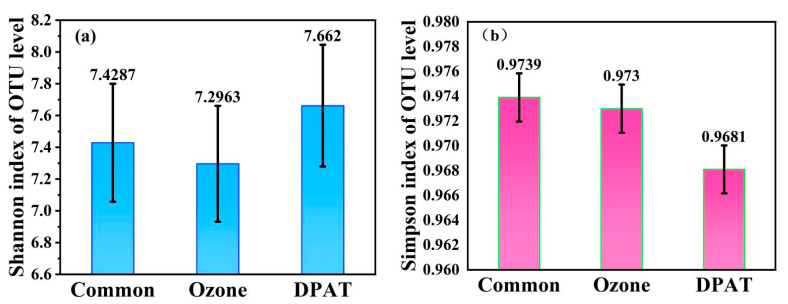
Alpha diversity of activated sludge samples evaluated by the Shannon index (**a**) and the Simpson index (**b**).

**Figure 7 molecules-27-05774-f007:**
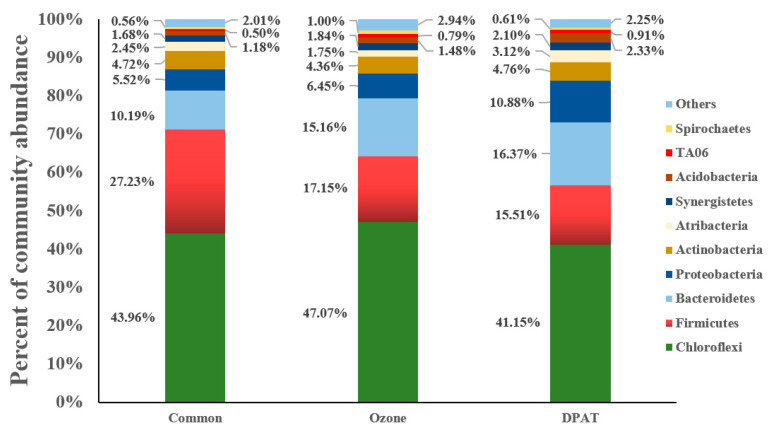
The community proportions in the sludge from common anaerobic treatment, ozone pretreatment, and DPAT treatment.

**Figure 8 molecules-27-05774-f008:**
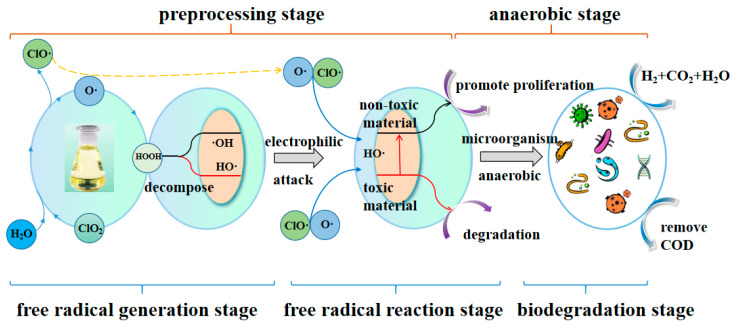
The reaction mechanisms of DPAT treatment.

**Table 1 molecules-27-05774-t001:** Initial physicochemical properties of experimental subjects.

Substance	Physical Properties	Chemical Properties
CMP wastewater	brown–black liquid	slightly alkaline
ozone-treated CMP wastewater	brown liquid	alkaline
anaerobic granular sludge	black granular	strong stability

**Table 2 molecules-27-05774-t002:** The compositions and related levels of organic contaminants in eucalyptus CMP wastewater before and after a chlorine dioxide pretreatment.

Organic Compound Name	Relative Level	Relative Level
**Before treatment:**		
Isobutyric acid	9.14%	_
Butyric acid	8.15%	_
2-Methyl-4-heptanone	0.35%	_
5-Methyl-5-propyl-nonane	0.44%	_
1-Iododecane	0.45%	_
2,6-di-tert-butyl-p-cresol	1.23%	_
3,5-Dimethyl-tert-butylbenzene	1.08%	_
2,2-Dichloro-1,1-difluoroethyl methyl ether	0.07%	_
1-iodooctadecane	0.69%	_
3,5-Dimethyl-4-octane	0.24%	_
o-Methyl-m-hydroxydiphenylamine	0.41%	_
Cyclohexanol	0.39%	_
4-Nitro-3-trifluoromethylphenol	0.19%	_
3-(2-Aminoethyl)indole	0.21%	_
Hexamide	0.82%	_
**After treatment:**		
3-octanone	_	0.23%
DL-Lipoamide	_	0.47%
L-lactide	_	2.39%

**Table 3 molecules-27-05774-t003:** The ratio between the COD mass and the mass of the applied chlorine dioxide.

C_CD_ (mg/L)	V_CD_ (mL)	M_CD_ (mg)	V_WW_ (mL)	C_COD_ (mg/L)	M_COD_ (mg)	M_COD_/M_CD_
465.41	20	9.31	300	4420	1326	142.46
	30	13.96		3983	1194.9	85.58
	40	18.62		3776	1132.8	60.85
	50	23.27		3966	1189.8	51.13
	60	27.92		4020	1206	43.19

**Table 4 molecules-27-05774-t004:** GC–MS results before and after anaerobic DPAT treatment.

Organic Compounds	Relative Content before Reaction	Relative Content after Reaction
3-octanone	0.23%	0.03%
DL-Lipoamide	0.47%	——
2-Methyl-2-butanol	——	0.61%
L-lactide	2.39%	0.09%
1,2-Propanediamine	——	0.15%

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
