# Peer review of "Developing an Efficient Processing System Treatment for the High Concentration of Eucalyptus Chemical Mechanical Pulp Wastewater"

_molecules, 2022, doi:10.3390/molecules27185774_

Round 1
Reviewer 1 Report
I recommend the publication after minor revision. The comments are described below pointed by line numbers.
At line 92 at the preparation of chlorine dioxide need to explain the chemical composition and difference between 2 reagents drug of A and drug of B.
Line 102 there are a mistake "tiosulfatethe" instead "thiosulfate the".
Line 116 2.5 Processing process is mentioned a comparison experiment by ozone oxidizer was carried out, but in the manuscript there are not described this experiment. Please describe the experiment.
Line 194 table 2 there are presented values for Cod Cod etc but there are not indicated the value for Mcd obtained from eq 3.
Line 283 figure 7 is not very clear the percent value for each type of microorganisms community. From figure result that others represent 100 percent; Spirochaetes 90% Firmicutes 70 percent. If on the color columns it introduced the percent value for each community it will be more relevant for understanding.
After that minor corrections will be made I recommend the publication of the manuscript.
Author Response
Response to the requires for modification
Thanks for the reviewers’ and the editors’ hard-work! According to the comments, we have revised the manuscript. The revisions are highlighted in yellow color.Please see the attachment
- At line 92 at the preparation of chlorine dioxide need to explain the chemical composition and difference between 2 reagents drug of A and drug of B.
Thanks for your comment. We have placed the relevant content on line 84 of the article.Drugs A and B for chlorine dioxide disinfectant were bought from Guangxi Aikening Disinfection Technology Co., Ltd.A is stable chlorine dioxide, B is a supporting activator, and the main component is acid.
- Line 102 there are a mistake "tiosulfatethe” instead "thiosulfate the".
Thanks for your comment. We made a correction on line 113 of the article.
- Line 116 2.5 Processing process is mentioned a comparison experiment by ozone oxidizer was carried out, but in the manuscript, there are not described this experiment. Please describe the experiment.
Thanks for your comment. We have added 2.5.2 to describe this experiment. Currently it's on line 130 of the article.
- Line 194 table 2 there are presented values for Cod Cod etc but there are not indicated the value for Mcd obtained from eq 3.
Thanks for your comment. We made corrections. The values ​​of the relevant parameters in Bulletin 3 can be observed in Table 3 on line 219 of the article.
- Line 283 figure 7 is not very clear the percent value for each type of microorganisms community. From figure result that others represent 100 percent; Spirochaetes 90% Firmicutes 70 percent. If on the color columns it introduced the percent value for each community it will be more relevant for understanding.
Thanks for your comment. We have annotated the content of Figure 7 so that the relevant data can be seen more clearly, which is convenient for analysis and comparison.

Reviewer 2 Report
Dear Editor
Thanks for inviting me to review this work. First this work sounds basically good, but there are lots of errors and also technical mistakes. I want to imply what and where but I found that I have to write several pages. So, I just highlighted (see MS file). Anyway, my decision is rejecting and I recommend this paper to be written well. The authors have to improve their paper in particular abstract, introduction and method etc. Also, they need a glossary for their abbreviation.

Author Response
Thanks for the reviewers’ and the editors’ hard-work! According to the comments, we have revised the manuscript. In terms of grammar, the English editing service provided by MDPI is used to ensure the accuracy of the description.The revisions are highlighted in yellow color.Please see the attachment.

Reviewer 3 Report
It looks interesting, but it needs improvement before publication.
1. Figure quality is not good. Please improve them. Letters in the figures are hardly readable.
2. Writing is not good. Please reorganize them and rewrite with more analysis.
3. The title is not suitable and does not represent its contents. Do not use abbreviation in the title. What is DPAT? Define it.
4. Please improve the graphical abstract. It does not deliver its core meaning and its quality is not good yet.
5. Microbial electrochemical systems (MFC, MEC, MDC, MRC, MESC) is an emerging wastewater treatment technology prouducing bioenergy. Please introduce them in the introduction or discussion.
MFC - Anode biofilm maturation time, stable cell performance time, and time-course electrochemistry in a single-chamber microbial fuel cell with a brush-anode. J Ind Eng Chem 2022 Feb; 106: 269-278
MFC - Power Generation and Anode Bacterial Community Compositions of Sediment Microbial Fuel Cells Differing in Anode Materials and Carbon Sources. Int J Electrochem Sci 2014 Jan; 9(1): 315-326
MEC - Microbial electrolysis cells for electromethanogenesis: Materials, configurations and operations. Environ Eng Res 2022 Feb; 27(1): 200484
6. There are unnecessary statements throughout the manuscript. Please rewrite and emphasize key findings.86. There is no critical discussion and key findings in this paper. Provide them.
Author Response
Thanks for the reviewers’ and the editors’ hard-work! According to the comments, we have revised the manuscript. The revisions are highlighted in yellow color.Please see the attachment.
- Figure quality is not good. Please improve them. Letters in the figures are hardly readable.Response:Thanks for your comment. We have adjusted all the images in the article so that they can be seen more clearly.
- Writing is not good. Please reorganize them and rewrite with more analysis.Response:Thanks for your comment. We have revised the article. And use MDPI's English editing service to make it even better.
- The title is not suitable and does not represent its contents. Do not use abbreviation in the title. What is DPAT? Define it.Thanks for your comment. We have adjusted the title. And define what is DPAT, they can be found on line 23 of the article.
- Please improve the graphical abstract. It does not deliver its core meaning and its quality is not good yet.Thanks for your comment. We have revised and adjusted the summary and summary graph. Make it more concise and intuitive.Because of formatting requirements, I put the abstract figure at the end of the article.
-
Microbial electrochemical systems (MFC, MEC, MDC, MRC, MESC) are an emerging bioenergy wastewater treatment technology. Please introduce them in an introduction or discussion. Thanks for your comment. We have added a reference to microbial electrochemical systems for wastewater treatment in the Introduction to enrich the content of the article. They can be found on line 56 of the article.
6. There are many unnecessary statements in the manuscript. Please rewrite and emphasize key findings. There are no critical discussions and key findings in this article. provide them. Thanks for your comment. We removed some unnecessary descriptions and added some key findings and key descriptions to the article. These are on lines 50, 81 and 350 of the article.

Reviewer 4 Report
The authors of the manuscript “Developing a DPAT treatment for the high concentration of eu calyptus CMP wastewater” conducted research on an important topic but they are unable to present it properly in a manuscript. The diagrams look good but the Methods section needs to be detailed and the results section requires critical scientific explanation for the obtained results in addition to proper citation to support the obtained results.
English and the grammar should be corrected throughout the manuscript.
Please write the full form of DPAT, CMP, DNBP, COD when it is first mentioned in the manuscript followed by the short form in brackets.
Necessary to highlight the novelty of the study
Insert a table in section 2.1 mentioning the initial physico-chemical characteristics of CMP wastewater, ozone-treated CMP wastewater, and anaerobic granular sludge
Please re-write section 2.1, very confusing and poor English.
What is drug A and B?
Was the method of preparation of Chlorine dioxide obtained from a previous literature? If yes cite reference.
Section 2.5.1. By how much amount chlorine dioxide disinfectant was added and for how long to how much amount of CMP wastewater?
Section 2.6 Mention the method used for COD/BOD and the instruments used.
Section 2.7.1 Mention the brand name of GCMS used.
Section 3.1 is missing. Please provide proper results with explanation.
Author Response
Response to the requires for modification
Thanks for the reviewers’ and the editors’ hard-work! According to the comments, we have revised the manuscript. The revisions are highlighted in yellow color.Please see the attachment
The authors of the manuscript "Development of a DPAT treatment method for high concentrations of eu calyptus CMP wastewater" investigated an important topic, but they were unable to present it correctly in the manuscript. The graphs look good, but the Methods section needs to be detailed and the Results section needs a critical scientific explanation of the results obtained, in addition to appropriate citations to support the results obtained.
response:
Thanks for your comment. We added 2.5.2 as a supplement to the experimental methods and adjusted the descriptions of other experimental methods to be more accurate. And adjusted the references, adding several references as support for the experimental results. In addition, we also improved 3.1 as a supplement to the experimental results to make the experimental results more convincing.
English and grammar should be corrected throughout the manuscript.
response:
Thanks for your comment. We have revised this article. And make it even better with MDPI's English editing services.
Please write the full form of DPAT, CMP, DNBP, COD when first mentioned in the manuscript and then fill in the short form in parentheses.
response:
Thanks for your comment. We've defined where they first appear. On lines 22, 23, 25.57.65.144 of the article, respectively.
It is necessary to highlight the novelty of the research
response:
Thanks for your comment. We supplement the description of the novelty of the article in Sections 344 and 3.6 of the article. And improved the description of the novelty of the article in the conclusion section
Insert a table in Section 2.1 mentioning the initial physicochemical properties of CMP wastewater, ozone treated CMP wastewater and anaerobic granular sludge
response:
Thanks for your comment. We added Table 1 in Section 2.1 to describe their initial physicochemical properties.
Please rewrite section 2.1, very confusing and bad English.
response:
Thanks for your comment. We have reorganized the language and improved it using MDPI's language editing service to ensure it is more accurate.
What are Drugs A and B?
response:
Thanks for your comment. We've put the relevant content on line 84 of the article. Drugs A and B are chlorine dioxide disinfectants, purchased from Guangxi Aikaining Disinfection Technology Co., Ltd. respectively, A is stable chlorine dioxide, B is a matching activator, and the main component is acid.
- Is the preparation method of chlorine dioxide from previous literature? If yes, please cite references.
response:
Thanks for your comment. We cited the relevant literature as the theoretical basis for the preparation method of chlorine dioxide.
- Section 2.5.1. How much chlorine dioxide disinfectant was added and how long was it added to how much CMP wastewater?
response:
Thanks for your comment. We have supplemented Section 2.5.1 to make it more complete. We used 300ml of CMP wastewater and performed gradient experiments by adding 20-60ml of chlorine dioxide to maximize efficiency and keep costs as low as possible. The reaction time for each sample was 30 minutes. A more specific description can be seen on line 126 of the article.
- Section 2.6 mentions the methods used for COD/BOD and the instruments used.
response:
Thanks for your comment. We supplemented the experimental setup. on line 145 of the article.
- Section 2.7.1 mentions the brand name of GCMS used.
response:
Thanks for your comment. We supplemented the GCMS brand name. on line 156 of the article.
- Missing Section 3.1. Please provide appropriate results and explain.
response:
Thanks for your comment. We supplement the experimental results in Section 3.1 to make it more specific and convincing.

Round 2
Reviewer 3 Report
It is not very satisfactory, but they improved it.